# Efficacy of an Online Educational Intervention in Reducing Body Weight in the Pre-Diabetic Population of 18–45 Years Old, a Randomized Trial Protocol

**DOI:** 10.3390/jpm12101669

**Published:** 2022-10-07

**Authors:** María del Valle Ramírez-Durán, Belinda Basilio-Fernández, Adela Gómez-Luque, Pilar Alfageme-García, María Zoraida Clavijo-Chamorro, Víctor Manuel Jiménez-Cano, Juan Fabregat-Fernández, Vicente Robles-Alonso, Sonia Hidalgo-Ruiz

**Affiliations:** 1Department of Nursing, University Center of Plasencia, University of Extremadura, 10600 Plasencia, Spain; 2Department of Nursing, Faculty of Nursing and Occupational Therapy, University of Extremadura, 10004 Cáceres, Spain

**Keywords:** diabetes mellitus type 2, body weight changes, clinical nursing research, nursing education research, exercise therapy, diet therapy, motivation

## Abstract

Aim: to analyze the efficacy of an educational online intervention focused on lifestyle changes in reducing body weight from baseline to 6 months in the pre-diabetic population of 18–45 years old in Extremadura (Spain). Methods: a single-blind, multicenter randomized parallel-comparison trial with two intervention groups in a 1:1 ratio will be carried out. Participants will be randomly assigned to intervention A or B with 37 cases in each group according to inclusion criteria of being enrolled or working at Extremadura University, scoring >7 points on the Findrisc test and not having diagnosed diabetes mellitus or physical disabilities. Intervention-A group will have access to online information about healthy diet and exercise. Intervention-B group will have access to a six-session educational program regarding behavioral changes in diet and exercise habits. They will complete follow-up activities and have a personal trainer and motivation. The primary outcome will be identifying changes in body weight from baseline to 1 and 6 months and between groups. The secondary outcomes will be accomplishing regular physical activity (>30 min/day or >4 h/week), decreasing sugary food intake or avoiding it altogether, increasing vegetable/fruit intake and lowering HbA1c levels to non-diabetic status when necessary.

## 1. Introduction

Diabetes mellitus (DM) represents one of the most prioritizing health issues due to the economic burden and social and sanitary impact [1]. It has been denominated as the ‘21st century epidemic’ as a result of the prevalence surge and the recurrent appearance of both chronic and acute complications associated with it [2,3]. Diabetes mellitus prevalence has been estimated at 9.3% worldwide and 10.5% in Spain among the 20–79 aged population by the International Diabetes Federation. Nowadays, 463 million people in the world are diagnosed with DM, 3.6 million in Spain. These data make Spain the fifth European country in the ranking of highest DM prevalence. By 2045, IDF estimates that there will have been an increase in the DM incidence of 700.2 million people in the world and 5 million in Spain [4].

Type-2 diabetes mellitus (T2DM) embodies 90–95% of all diabetes diagnoses in developed countries. On this account, modifiable factors such as obesity and unhealthy lifestyles are associated with the increase in type-2 diabetes, triggering precocious T2DM development among children and adolescents [1,5].

Diabetes screenings are performed in the presence of overweight/obesity or other risk factors such as family DM history, race (African, African American, Hispanic, Native Americans, Asians, and Pacific islanders), cardiovascular alterations, hypertension, polycystic ovary, dyslipidemias, sedentary lifestyle, HIV or being older than 45 years old [5,6,7]. Notwithstanding this fact, undiagnosed diabetes rates are 40.7%, thus being more prone to developing complications [4].

T2DM management focuses not only on diagnosis, but the literature shows that changes in the lifestyle can prevent its advance, being also cost-effective [7,8,9,10,11,12]. Consequently, early screening interventions to detect at-risk populations are paramount, especially in those younger than 45 years old where the incidence is increasing and individuals usually are not a target in the healthcare screening process [13].

The T2DM diagnosis gold standard is the oral glucose tolerance test (OGTT) which contains drawbacks of being a complex, burdensome and expensive test [11,14,15]. Nonetheless, the Finnish Diabetes Risk Score (FINDRISC) has been proven as a cost-effective alternative that allows clinicians to predict T2DM risks and screen for prediabetes status. It assesses age, body mass index (BMI), waist circumference, hypertension history, regular physical activity, daily consumption of fruits and vegetables, and DM family history [7,9,10,11]. Several large studies in Europe have employed FINDRISC as an efficient tool to screen diabetes such as the German National Diabetes Prevention Programme [16], the Diabetes in Europe: Prevention using Lifestyle, Physical Activity and Nutrition intervention (DEPLAN) [17] or the European Guideline and Training Standards for Diabetes Prevention (IMAGE) [18].

The American Diabetes Association (ADA) indicates that changes toward a healthier diet and an active lifestyle—avoiding overweight—throughout educational interventions have been proven effective, with a positive impact on T2DM prevention [6]. In addition, several randomized trials testing prevention programs focused on education have also been effective. For instance, the Diabetes Prevention Program (DPP) improved anthropometric and certain metabolic outcomes, reducing by 58% the T2DM incidence in 3 years [8]. In the Finnish Diabetes Prevention Study, T2DM incidence was lowered by 43% after 7 years [19]. The Da Quin study reduced the T2DM incidence by 39% after 30 years [20]. Finally, the metanalysis carried out by Renders et al. analyzed 41 studies, concluding that educational interventions have a positive effect on reducing T2DM [21].

Endpoint variation is shown in educational-centered studies by either focusing on changes in HbA1c values, diet, or body weight. Notwithstanding that fact, effectiveness is shown [22,23,24,25]. However, studies focused on younger populations are lacking, especially among 18–45 years old. Furthermore, the COVID-19 pandemic rescinded all health group-education interventions or activities and hindered individual education as well, being historically scarce and underdeveloped [26]. However, new technologies have proven their efficacy as an educational tool in DM, being a new resource to exploit in order to understand which educational interventions are the most efficient [27,28,29].

The study hypothesis is that online educational intervention focused on lifestyle changes is effective in reducing body weight when engaging activities are encouraged and present throughout the first weeks of behavioral modifications.

The aim of the study is to analyze the efficacy of an online educational intervention focused on lifestyle changes in reducing body weight from baseline to 1 and 6 months in the pre-diabetic population of 18–45 years old in Extremadura (Spain).

## 2. Materials and Methods

### 2.1. Study Design and Settings

The study is a single-blind, multicenter randomized parallel-comparison trial with two intervention groups in a 1:1 ratio. Participants were blinded to the allocated group, but researchers performing the intervention were not.

The trial registration code is 10.17605/OSF.IO/4D8FJ and was designed so that it can be reported according to the SPIRIT 2013 statement [30]. The study will be carried out across three centers of the University of Extremadura (UEx).

### 2.2. Participants: Eligibility, Recruitment, and Allocation

Participants were eligible for this trial when in accordance with the following selection criteria:

As inclusion criteria: ages 18–45, FINDRISC screening result >7 points, being enrolled or working at the University of Extremadura. As exclusion criteria: HbA1c compatible with type 2 diabetes, diabetes diagnosis, mobility disability and/or comorbidities incompatible with moderate exercise. 

### 2.3. Sample Size

The primary outcome was body weight change at 1 and 6 months. The power calculation assumed a 14 kg SD in baseline weight and a 0.9 correlation between baseline and 6-month weight [31]. As such, a sample size of 74 (37 per group) was required to give the trial 80% power to detect a 4 kg difference in mean weight change between groups at 6 months (*p* < 0.05, two-sided test).

### 2.4. Recruitment

All participants from 18 to 45 years old will be reached through a standardized email sent to their university’s account where they will find an explanation of the project and the informed consent form with the possibility of either accepting to participate or declining any participation. Five researchers will reinforce the alumni’s participation with a visit during class hours and welcome to the project and explain it further. In the case of the university’s workforce, these five researchers will contact them during work hours in their workplace.

After two weeks’ notice, all volunteers who had signed the informed consent form will be summoned to follow the screening path which will be carried out using the Spanish version of the FINDRISC, which is a validated eight-item European questionnaire related to diabetes risk factors. This questionnaire has been validated in Spanish by the Pizarra Study in a Spanish population aged 18 to 65 years old. The results showed a 22.2% positive predictive value and 96% negative [11]. The study DE-PLAN—also developed in Spain—showed a 75.9% sensitivity and 52.3% T2DM diagnosis specificity and 65.8% prediabetes diagnosis specificity [32,33]. Furthermore, its utility has been demonstrated all over the world [16,32,33,34,35].

The FINDRISC screening will be carried out by six research nurses at the university centers. To those eligible participants with Findrisc results >7 points, researchers will ask them whether they have any mobility disability and/or comorbidities incompatible with moderate exercise in order to exclude those participants who do not meet the inclusion/exclusion criteria.

All suitable participants will be informed about their prediabetes status. Consequently, they will be offered to enter the study and will be given information and the calendar for the study.

As a result of participating in the study, additional clinical data will be collected such as capillary blood samples to measure HbA1c and fasting plasma glucose, lipid profile, BMI and blood pressure.

Furthermore, participants will fill in four questionnaires in the Spanish version regarding current exercise and diet preferences, and diabetes knowledge. To assess exercise, we will use the Rapid Assessment of Physical Activity questionnaire (RAPA) [36]. To assess participants’ current diet, we will use the Three-Factor Eating Questionnaire (Tfeq-Sp) [37], Yale Food Addiction (YFAS) [38] and Emotional Eater Questionnaire (EEQ) [39]. Finally, to assess diabetes knowledge, we will use the Diabetes Knowledge Questionnaire (DKQ-24) [40]. Socioeconomic data will also be collected.

The recruitment will be carried out first in the Cáceres UEx center, followed by the UEx center in Mérida and Badajoz. The researchers will be the same in all centers, and it will be executed from September 2022 to April 2024.

### 2.5. Allocation

Participants will be randomly allocated to either intervention A or intervention B group by one researcher using a randomization list. The participant timeline can be read in Figure 1.

### 2.6. Intervention

Both groups will be separately summoned to receive details about the intervention protocol in a welcome synchronous session at the university.

Participants allocated to the intervention-A group will have been provided access to information about healthy diet and exercise through the university’s online campus, to help them implement long-term changes in their diet and exercise habits using videos, recorded classes, and online information. A reminder of the following measurement appointment in one month will be sent and then for the 6 months final data collection. This intervention group will not be encouraged or motivated by the researchers but rather informed of their status and recommended to implement healthy diet and exercise habits.

Participants allocated to the intervention B group will have been provided with a six-session educational program mixing synchronous and asynchronous methodology concerning educational and behavioral changes regarding diet and exercise habits. All sessions will be carried out on the university’s online campus in one month so participants can promptly implement their exercise routines and diet changes through several proposed activities that they should complete. The first month of lifestyle treatment is a critical period for helping participants achieve weight loss according to the results of Miller et al. [41].

Participants will be assigned a personal trainer to make follow-ups and provide personal motivation. This activity will be held by two clinical nurse researchers who are trained in motivating patients. There will be constant feedback regarding the process and possible distractors via mail, telephone, or videoconference in order to surge adherence to the intervention.

Each session will have 2 h duration and pivot on three main issues: diet, exercise, and motivation, which according to Zhang et al. have been proven to be the central issues to work on [29]. Using synchronous methods will help promote participant retention and complete follow-up.

The first session, named “walking toward a healthy diet”, will be dedicated to identifying those fatty and sugary foods (both hidden and naturally present) and explaining strategies to avoid them by interchanging them for healthier options.

It will also help participants acknowledge sources of stress linked to binge eating and provide them with some strategies according to the scores in the questionnaires. Furthermore, participants will learn about type-2 diabetes through an educational video created by nurse researchers.

After this session, participants will begin to register their fruits and vegetables and high fat and sugar food intake in a diary. Furthermore, they will try to identify barriers and stressors to a healthy diet.

In the second session, named “give exercise the value that it has”, we will focus on explaining the benefits of exercising, showing participants easy ways of exercising, and how to improve exercise adherence using motivation.

After this session, participants will begin to implement strength and aerobic exercise in their routine and register it using fitness tracker wristbands.

Session three, named “barriers and stressors that block me from implementing a healthy diet”, will be a synchronous group session focused on those found barriers and stressors to maintaining a healthy diet and designing a plan to tackle them. After this session, participants will implement the plan and register any digression.

Session four, named “get up and get moving”, will help reinforce exercise habits and give participants the option to discuss any encountered limitations, their best exercise hours, and exercise planning. After this session, participants will further implement their exercise plan.

Session five, named “dealing with our surroundings I”, will be a synchronous group session focused on a real-life testimonial speech from a young diabetic person who will discuss what it feels like to deal with type-2-diabetes at a young age. A colloquial will be initiated after the speech for every participant to ask or discuss any issue. Additionally, participants will be asked to share a healthy picnic with friends or family and post photos on the online campus. Another option will be to post on the online campus a video of themselves preparing a healthy meal. The purposes will be overcoming negative social signals related to food and being able to make healthy decisions regarding “eating outside”.

Session six, named “dealing with our surroundings II”, will be an asynchronous session showing Spanish influencers who promote healthy diets and exercises. After the session, participants will share photos of themselves practicing any exercise.

After these sessions, participants will be able to contact the researchers via email, telephone, or videoconference when necessary.

### 2.7. Outcomes of Interest

#### 2.7.1. Primary Outcome

The primary outcome will be identifying changes in body weight from baseline to 1 and 6 months and between groups. To measure body weight and BMI, we will use TANITA MC-780MAS Segmental. In the welcome session, participants will be presented with the primary outcome, which is maintaining a healthy weight or losing 5% if overweight.

#### 2.7.2. Secondary Outcomes

All secondary outcomes will be assessed at baseline, 1 and 6 months after the intervention.

A secondary outcome will be accomplishing regular physical activity (>30 min/day or >4 h/week). To assess physical activity, participants will complete the RAPA questionnaire. This questionnaire in the Spanish version obtained good test–retest reliability (r = 0.906), a sensitivity of 76.6% and a specificity of 65.6% [36].

Another secondary outcome will be decreasing sugary food intake or avoiding it altogether and increasing vegetable/fruit intake. We will use the Tfeq-Sp, Yale food addiction and EEQ to assess current diet and willingness to change it. The Tfeq-Sp measures three concepts: cognitive restraint, emotional eating, and uncontrolled eating. These concepts are explained further in Jauregui-Lobera et al. [37]. The internal consistency of the Tfeq-Sp was determined by means of Cronbach’s α coefficient, with values ranging between 0.75 and 0.87 [37]. The Yale food addiction questionnaire identifies individuals with addictive behaviors toward food in accordance with substance dependence criteria. The validated Spanish version demonstrated internal consistency for the one single dimension solution of α = 0.95 [38]. The EEQ measures three factors: disinhibition, high-calorie food preference, and feelings of guilt. EEQ internal consistency was good (α = 0.753) [39].

Lowering HbA1c levels to non-diabetic status, when necessary, will also be a secondary outcome. To measure HbA1c, we will employ the Affinity A1c Analyzer.

Finally, changes in diabetes knowledge will be measured using the DKQ-24 questionnaire. This questionnaire was designed to assess overall diabetes knowledge according to content recommendations in the National Standards for Diabetes Patient Education Programs. The 24-item version of the DKQ-24 attained a reliability coefficient of 0.78, indicating internal consistency [40].

### 2.8. Data Analysis

#### 2.8.1. Statistical Analysis

Descriptive statistics will be estimated for baseline patients and their characteristics. The similarity between groups at baseline will be calculated, followed by examining differences between groups in body weight. Covariates will be screened in bivariate analyses and included in multivariable analysis when related to outcomes *p* < 0.2. Covariates will include age, sex, center, BMI, HbA1c, lipid profile and blood pressure. Adjustment for confounding factors will be examined using linear regression. Questionnaire scores will be assessed, and changes will be sought between baseline, 1 and 6 months and between groups. All statistical analyses will be performed using IBM SPSS statistics version 27.0.

#### 2.8.2. Data Management

Personal information obtained in this study will be coded and anonymized.

All the data for the present study will be managed securely for 5 years from completion. All materials will be disposed of after this period, and sufficient care will be taken to ensure that the names of individuals on the signed informed consent forms cannot be identified.

#### 2.8.3. Criteria for Discontinuation of Participation in the Study or of the Study

If a participant withdraws consent for participation in the study.If their physician determines that the continuation of the intervention is not preferable for an individual patient. At the moment, there is no reason that can be used to exclude a patient from the trial apart from the aforementioned exclusion criteria, which might minimize bias affecting the results of the study.If the study protocol cannot be followed.

### 2.9. Monitoring

Information about the initiation of the study, the conduct of the study, ethical considerations, the occurrence of detrimental or adverse events, the results of the study, and registration of the study with a public database will be submitted to our ethics committee.

#### 2.9.1. Protocol Amendments

If any amendments are made, the Ethics Committee will be notified.

#### 2.9.2. Follow-Up of Adverse Events

This study involves the implementation of a lifestyle program focused on a healthy diet and exercise to help persons with prediabetes reduce their risks to develop type-2 diabetes.

There is little likelihood of any health hazards. If any serious adverse events do occur, they will be reported in line with the standard operating procedure for Reporting Serious Adverse Events in Clinical Research.

### 2.10. Ethics and Dissemination

The study is implemented in accordance with the Declaration of Helsinki and the ethical guidelines for medical research covering humans. This study has been approved by the Ethical Committee of Extremadura University (No.165//2021). No information identifying specific patients is stored alongside study data. Only the authors will have access to the final trial dataset. It is considered that harm to the patient will be rare because the examinations performed will be those undertaken in the usual medical examination. However, participants will freely exit the study if they want to.

## 3. Discussion

The striking increase in the incidence of type-2 diabetes mellitus and its broad consequences has led us to focus on preventing its development through educational interventions that reduce the associated risk regarding lifestyle choices [22,23,24,25].

However, studies in our target population are scarce mostly due to lower incidence compared with other age ranges. Notwithstanding that fact, given the chronic character that diabetes mellitus has, it is paramount to prevent its development at any age range. As a result, we chose to run a parallel-comparison trial so every patient—regardless of their allocation in the study—will be aware of their pre-diabetic status and make evidence-based and healthy changes in their lifestyle.

COVID-19 has strained and changed the healthcare system during these past two years, being a priority over chronic diseases [42]. Furthermore, both group and individual interventions onsite were suspended. However, telemedicine has increased and developed greatly, becoming an effective tool to deliver educational interventions. There have been research studies prior to and during COVID-19 that have proven the efficacy of online educational interventions [27,28,29]. As a result, these interventions have the potential to bridge the gap between the limited supply of healthcare resources and the growing demand for diabetes education.

There are several potential limitations of the planned study. First, the assessment of body weight changes from baseline to 6 months might reflect only those changes occurred in the short term. This will be taken into account in further research. Nonetheless, several studies have set up 6 months to collect endpoint assessments [28,31].

Second, even though all university staff will be included in the study, that is, low, middle and high educational status population, given the highest high-educational status population rate, the low and middle-educational status population might be underrepresented.

Third, by carrying out a parallel-comparison trial with two intervention groups, we expect changes in both groups. However, our study will assess whether follow-up and motivation are needed in achieving lifestyle changes, providing a viewpoint that could help healthcare professionals manage their workload more efficiently.

As a consequence, we expect to find an improvement leveling off those T2MD risk factors in the pre-diabetic population aged 18–45 years old, hypothesizing that follow-up and motivation are needed to achieve lifestyle changes when using online educational intervention focused on improving diet and exercise lifestyle choices.

## Figures and Tables

**Figure 1 jpm-12-01669-f001:**
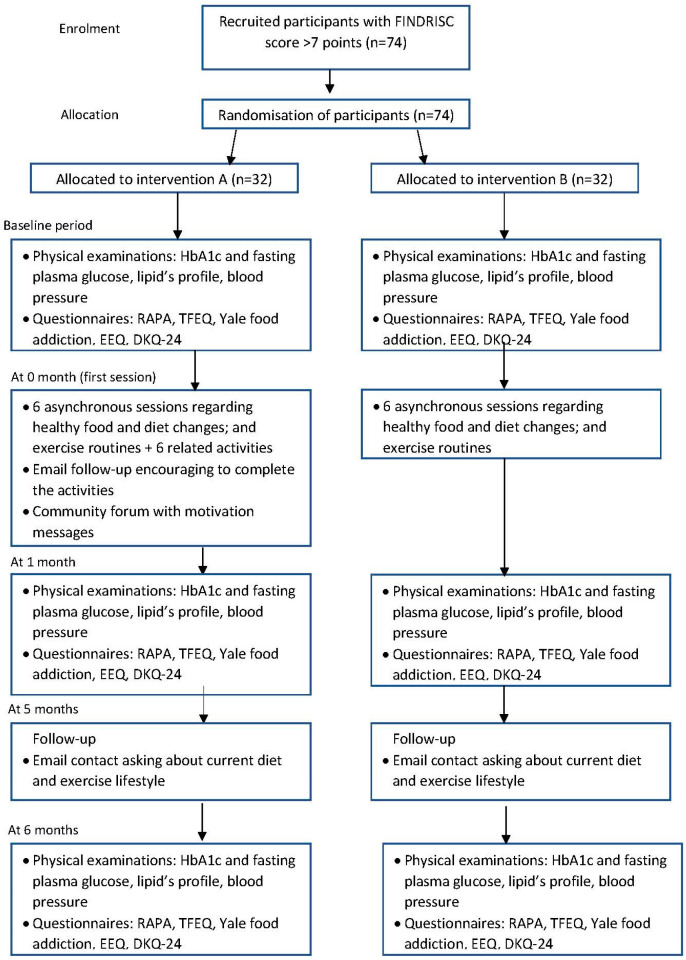
Participant flow diagram.

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
