# Peer review of "Efficacy of an Online Educational Intervention in Reducing Body Weight in the Pre-Diabetic Population of 18–45 Years Old, a Randomized Trial Protocol"

_jpm, 2022, doi:10.3390/jpm12101669_

Round 1
Reviewer 1 Report
An interesting study but the subjects to be studied are either university students or university employes thus of high educational status.. A worry that the control group will also change their behaviour.A 6 month study in the lifetime of patients with only diabetes risk seems short.
Reviewer 2 Report
The study is very interesting. However, I do not understand if the study is complete or if it is still being processed. The author has no tables or clear statistics that show their work.
The discussion part of the study needs to be rewritten. Weaknesses and strengths of the study must be clearly stated.
Reviewer 3 Report
The authors did a commendable job compiling this protocol. This protocol helps in preventing Diabetes Mellitus through educational interventions which help reduce the associated risk regarding lifestyle choices. The authors propose to run a parallel-comparison of trial so every patient-regardless of their allocation in the study, the patients will be aware of their prediabetic status and make evidence directed healthy changes in their lifestyle. Telemedicine interventions have the potential to address the glut between the limited supply of healthcare resources and the growing demand for diabetes education. The study proposed in the protocol will assess the impact of follow-up and motivation in achieving lifestyle changes, and how that could help healthcare professionals manage their workload in an effective way.
